# THE IMPORTANCE OF THE CURRENT INPUT IN SEQUENCE MODELING

## ABSTRACT

The last advances in sequence modeling are mainly based on deep learning approaches. The current state of the art involves the use of variations of the standard LSTM architecture, combined with several tricks that improve the final prediction rates of the trained neural networks. However, in some cases, these adaptations might be too much tuned to the particular problems being addressed. In this article, we show that a very simple idea, to add a direct connection between the input and the output, skipping the recurrent module, leads to an increase of the prediction accuracy in sequence modeling problems related to natural language processing. Experiments carried out on different problems show that the addition of this kind of connection to a recurrent network always improves the results, regardless of the architecture and training-specific details. When this idea is introduced into the models that lead the field, the resulting networks achieve a new state-of-the-art perplexity in language modeling problems.

## 1 INTRODUCTION

Deep learning models constitute the current state of the art in most artificial intelligence applications, from computer vision to robotics or medicine. When dealing with sequential data, Recurrent Neural Networks (RNNs), specially those architectures with gating mechanisms such as the LSTM (Hochreiter & Schmidhuber, 1997), the GRU (Cho et al., 2014) and other variants, are usually the default choice. One of the most interesting applications of RNNs is related to the field of Natural Language Processing, where most tasks, such as machine translation, document summarization or language modeling, involve the manipulation of sequences of textual data. Of these, language modeling has been extensively used to test different innovations in recurrent architectures, mainly due to the ease of obtaining very large datasets that can be used to train neural networks with millions of parameters.

Sequence modeling consists of predicting the next element in a sequence given the past history. In language modeling, the sequence is a text, and hence the task is to predict the next word or the next character. In this context, some of the best performing architectures include the Mogrifier LSTM (Melis et al., 2020) and different variations of the AWD-LSTM (Merity et al., 2018), usually combined with dynamic evaluation and mixture of sofmaxes (MoS) (Wang et al., 2019; Gong et al., 2018). These models obtain the best state-of-the-art performance with moderate size datasets, such as the Penn Treebank (Mikolov et al., 2010) or the Wikitext-2 (Merity et al., 2017) corpora, when no additional data are used during training. When larger datasets are considered, or when external data are used to pre-train the networks, attention-based architectures usually outperform other models (Radford et al., 2019; Brown et al., 2020).

In this work we use moderate-scale language modeling datasets to explore the effect of a mechanism recently proposed by Oliva & Lago-Fernández (2021), when combined with different LSTM-based models in the language modeling context. The idea consists of modifying a recurrent architecture by introducing a direct connection between the input and the output of the recurrent module. This has been shown to improve both the model's generalization results and its readability in simple tasks related to the recognition of regular languages.

In a standard RNN, the output depends only on the network's hidden state, $h_t$, which in turn depends on both the input, $x_t$, and the recent past, $h_{t-1}$. But there is no explicit dependence of the network's

output on its input. In some cases this could be a shortcoming, since the transformation of $x_t$ needed to compute the network's internal state is not necessarily the most appropriate to compute the output. However, an explicit dependence of the output on $x_t$ can be forced by adding a *dual* connection that skips the recurrent layers. We claim that this strategy may be of general application in RNN models.

To test our hypothesis we perform a thorough comparison of several state-of-the-art RNN architectures, with and without the *dual* connection, on the Penn Treebank and the Wikitext-2 datasets. Our results show that, under all experimental conditions, the *dual* architectures outperform their non-dual counterparts. In addition, the Mogrifier-LSTM enhanced with a *dual* connection establishes a new state-of-the-art word-level perplexity for the Penn Treebank dataset when no additional data are used to train the models.

The remainder of the article is organized as follows. First, in section 2, we present the different models we have used and the two possible architectures, the standard recurrent architecture and the *dual* architecture. In section 3, we describe the datasets and the experimental setup. In section 4, we present our results. And finally, in section 5, we extract some conclusions and discuss further lines of research.

## 2 MODELS

We start by presenting the standard recurrent architecture which is common to all the models. In absence of a *dual* connection, the basic architecture involves an embedding layer, a recurrent layer and a fully-connected layer with $softmax$ activation:

$$
\begin{align}
\boldsymbol{e}_t &= \boldsymbol{W}^{ex}\boldsymbol{x}_t \tag{1}\\
\boldsymbol{h}_t &= REC(\boldsymbol{e}_t, \mathbb{S}_{t-1}) \tag{2}\\
\boldsymbol{y}_t &= softmax(\boldsymbol{W}^{yh}\boldsymbol{h}_t + \boldsymbol{b}^y), \tag{3}
\end{align}
$$

where $\boldsymbol{W}^{**}$ and $\boldsymbol{b}^*$ are weight matrices and biases, respectively, and $\boldsymbol{x}_t$ is the input vector at time $t$. The $REC$ module represents an arbitrary recurrent layer, with $\mathbb{S}_{t-1}$ being a set of vectors describing its internal state at the previous time step. In the most general case, this module will simply be an LSTM cell, but we consider other possibilities as well, as described below.

The *dual* architecture introduces an additional layer, with ReLU activation, which is fed with both the output of the embedding layer and the output of the recurrent module:

$$
\begin{align}
\boldsymbol{e}_t &= \boldsymbol{W}^{ex}\boldsymbol{x}_t \tag{4}\\
\boldsymbol{h}_t &= REC(\boldsymbol{e}_t, \mathbb{S}_{t-1}) \tag{5}\\
\boldsymbol{d}_t &= ReLU(\boldsymbol{W}^{de}\boldsymbol{e}_t + \boldsymbol{W}^{dh}\boldsymbol{h}_t + \boldsymbol{b}^d) \tag{6}\\
\boldsymbol{y}_t &= softmax(\boldsymbol{W}^{yd}\boldsymbol{d}_t + \boldsymbol{b}^y). \tag{7}
\end{align}
$$

This way the network's input can reach the softmax layer following two different paths, through the recurrent layer and through the *dual* connection. In the following we consider different forms for the recurrent module in equations 2 and 5.

### 2.1 THE LSTM MODULE

In the simplest approach the recurrent module consists of an LSTM cell, where the internal state includes both the output and the memory, $\mathbb{S}_t = \{\boldsymbol{h}_t; \boldsymbol{c}_t\}$, which are computed as follows:

$$
\begin{align}
\boldsymbol{f}_t &= \sigma(\boldsymbol{W}^{fe}\boldsymbol{e}_t + \boldsymbol{W}^{fh}\boldsymbol{h}_{t-1} + \boldsymbol{b}^f) \tag{8} \\
\boldsymbol{i}_t &= \sigma(\boldsymbol{W}^{ie}\boldsymbol{e}_t + \boldsymbol{W}^{ih}\boldsymbol{h}_{t-1} + \boldsymbol{b}^i) \tag{9} \\
\boldsymbol{o}_t &= \sigma(\boldsymbol{W}^{oe}\boldsymbol{e}_t + \boldsymbol{W}^{oh}\boldsymbol{h}_{t-1} + \boldsymbol{b}^o) \tag{10} \\
\boldsymbol{z}_t &= tanh(\boldsymbol{W}^{ze}\boldsymbol{e}_t + \boldsymbol{W}^{zh}\boldsymbol{h}_{t-1} + \boldsymbol{b}^z) \tag{11} \\
\boldsymbol{c}_t &= \boldsymbol{f}_t \odot \boldsymbol{c}_{t-1} + \boldsymbol{i}_t \odot \boldsymbol{z}_t \tag{12} \\
\boldsymbol{h}_t &= \boldsymbol{o}_t \odot tanh(\boldsymbol{c}_t), \tag{13}
\end{align}
$$

where, as before, $\boldsymbol{W}^{**}$ are weight matrices and $\boldsymbol{b}^*$ are bias vectors. The $\odot$ operator denotes an element-wise product, and $\sigma$ is the logistic sigmoid function. For convenience, we summarize the joint effect of equations 8-13 as:

$$
\boldsymbol{h}_t = LSTM(\boldsymbol{e}_t, \{\boldsymbol{h}_{t-1}; \boldsymbol{c}_{t-1}\}). \tag{14}
$$

In the literature it is quite common to stack several LSTM layers. Here we consider a double-layer LSTM, where the output $\boldsymbol{h}_t$ of the recurrent module is obtained by the concatenated application of two LSTM layers:

$$
\begin{align}
\boldsymbol{h}'_t &= LSTM_1(\boldsymbol{e}_t, \{\boldsymbol{h}'_{t-1}; \boldsymbol{c}'_{t-1}\}) \tag{15} \\
\boldsymbol{h}_t &= LSTM_2(\boldsymbol{h}'_t, \{\boldsymbol{h}_{t-1}; \boldsymbol{c}_{t-1}\}). \tag{16}
\end{align}
$$

We refer to this double LSTM module as $dLSTM$:

$$
\begin{align}
\boldsymbol{h}_t &= dLSTM(\boldsymbol{e}_t, \{\boldsymbol{h}_{t-1}; \boldsymbol{c}_{t-1}; \boldsymbol{h}'_{t-1}; \boldsymbol{c}'_{t-1}\}) \tag{17} \\
&= LSTM_2(LSTM_1(\boldsymbol{e}_t, \{\boldsymbol{h}'_{t-1}; \boldsymbol{c}'_{t-1}\}), \{\boldsymbol{h}_{t-1}; \boldsymbol{c}_{t-1}\}). \tag{18}
\end{align}
$$

## 2.2 THE MOGRIFIER-LSTM MODULE

The Mogrifier-LSTM (Melis et al., 2020) is one of the state-of-the-art variations of the standard LSTM architecture achieving the lowest perplexity scores in language modeling tasks. It basically consists of a standard LSTM block, but the input $\boldsymbol{e}_t$ and the hidden state $\boldsymbol{h}_{t-1}$ are transformed before entering equations 8-13. The mogrifier transformation involves several steps where $\boldsymbol{e}_t$ and $\boldsymbol{h}_{t-1}$ modulate each other:

$$
\begin{align}
\boldsymbol{e}_t^i &= 2\sigma(\boldsymbol{Q}^i \boldsymbol{h}_{t-1}^{i-1}) \odot \boldsymbol{e}_t^{i-2}, \quad \text{for odd } i \in \{1, 2, ..., r\} \tag{19} \\
\boldsymbol{h}_{t-1}^i &= 2\sigma(\boldsymbol{R}^i \boldsymbol{e}_t^{i-1}) \odot \boldsymbol{h}_{t-1}^{i-2}, \quad \text{for even } i \in \{1, 2, ..., r\}, \tag{20}
\end{align}
$$

where $\boldsymbol{Q}^i$ and $\boldsymbol{R}^i$ are weight matrices and we have $\boldsymbol{e}_t^{-1} = \boldsymbol{e}_t$ and $\boldsymbol{h}_{t-1}^0 = \boldsymbol{h}_{t-1}$. The linear transformations $\boldsymbol{Q}^i \boldsymbol{h}_{t-1}^{i-1}$ and $\boldsymbol{R}^i \boldsymbol{e}_t^{i-1}$ can also include the addition of a bias vector, which has been omitted for the sake of clarity. The constant $r$ is a hyperparameter whose value defines the number of rounds of the transformation. We refer to this recurrent module, including the mogrifier transformation and the subsequent application of the LSTM layer, as:

$$
\boldsymbol{h}_t = mLSTM(\boldsymbol{e}_t, \{\boldsymbol{h}_{t-1}; \boldsymbol{c}_{t-1}\}) = LSTM(\boldsymbol{e}_t^*, \{\boldsymbol{h}_{t-1}^*; \boldsymbol{c}_{t-1}\}), \tag{21}
$$

where $\boldsymbol{e}_t^*$ and $\boldsymbol{h}_{t-1}^*$ are the highest indexed $\boldsymbol{e}_t^i$ and $\boldsymbol{h}_{t-1}^i$ in equations 19 and 20. Note that the choice $r = 0$ recovers the standard LSTM model.

Melis et al. (2020) also used a double-layer LSTM enhanced with the mogrifier transformation. This strategy can be summarized as follows:

$$\begin{aligned}
\boldsymbol{h}_t &= mdLSTM(\boldsymbol{e}_t, \{\boldsymbol{h}_{t-1}; \boldsymbol{c}_{t-1}; \boldsymbol{h}'_{t-1}; \boldsymbol{c}'_{t-1}\}) & (22)\\
&= mLSTM_2(mLSTM_1(\boldsymbol{e}_t, \{\boldsymbol{h}'_{t-1}; \boldsymbol{c}'_{t-1}\}), \{\boldsymbol{h}_{t-1}; \boldsymbol{c}_{t-1}\}). & (23)
\end{aligned}$$

## 3 EXPERIMENTS

### 3.1 DATASETS

We perform experiments on two datasets: the Penn Treebank corpus (Marcus et al., 1993), as pre-processed by Mikolov et al. (2010), and the WikiText-2 dataset (Merity et al., 2017). In both cases, the data are used without any additional preprocessing.

The Penn Treebank (PTB) dataset has been widely used in the literature to experiment with language modeling. The standard data preprocessing is due to Mikolov et al. (2010), and includes transformation of all letters to lower case, elimination of punctuation symbols, and replacement of all numbers with a special token. The vocabulary is limited to the 10,000 most frequent words. The data is split into a training set which contains almost 930,000 tokens, and validation and test sets with around 80,000 words each.

The WikiText-2 (WT2) dataset, introduced by Merity et al. (2017), is a more realistic benchmark for language modeling tasks. It consists of more than 2 million words extracted from Wikipedia articles. The training, validation and test sets contain around 2,125,000, 220,000, and 250,000 words, respectively. The vocabulary includes over 30,000 words, and the data retain capitalization, punctuation, and numbers.

### 3.2 EXPERIMENTAL SETUP

All the considered models follow one of the two architectures discussed in section 2, either the Embedding-Recurrent-Softmax (ERS) architecture (equations 1-3) or the *dual* architecture (equations 4-7). In either case, the recurrent module can be any of $LSTM$, $dLSTM$, or $mdLSTM$. Weight tying (Inan et al., 2017; Press & Wolf, 2017) is used to couple the weight matrices of the embedding and the output layers. This reduces the number of parameters and prevents the model from learning a one-to-one correspondence between the input and the output (Merity et al., 2018).

We run two different sets of experiments. First, we analyze the effect of the *dual* connection by comparing the performances of the two architectures (ERS vs Dual), using each of the recurrent modules, on both the PTB and the WT2 datasets. In this setting the hyperparameters are tuned for the ERS architecture, and then transferred to the *dual* case. Second, we search for the best hyperparameters for the *dual* architecture using the $mdLSTM$ recurrence, and compare the perplexity score with current state-of-the-art values. All the experiments have been performed using the Keras library (Chollet et al., 2015), and the implementation is available in a public Github repository[1].

The networks are trained using the Nadam optimizer (Dozat, 2016), a variation of Adam (Kingma & Ba, 2015) where Nesterov momentum is applied. The number of training epochs is different for each experimental condition. On one hand, when the objective is to perform a pairwise comparison between *dual* and non-dual architectures, we train the models for 100 epochs. On the other hand, when the goal is to compare the *dual* network with state of the art approaches, we let the models run for 300 epochs. We use batch sizes of 32 and 128 for the PTB and the WT2 problems, respectively, and set the sequence length to 25 in all cases. The remaining hyperparameters are searched in the ranges described in table 1.

Finally, all the models are run twice, both with and without dynamic evaluation (Krause et al., 2018). Dynamic evaluation is a standard method commonly used to adapt the model parameters, learned during training, using also the validation data. This allows the networks to get adapted to the new evaluation conditions, which in general improves their performance. In order to keep the models as simple as possible, no additional modifications have been considered.

---

[1]The repository is not yet available to preserve author anonymity. It will be released after the review process.

Table 1: List of all the hyperparameters and the search range associated with each of them. Those marked with an asterisk (*) refer to the *dual* architectures only.

| Name | Description | Values |
|------|-------------|--------|
| *Num epochs* | Number of training epochs. | $\{100, 300\}$ |
| *Learning rate* | Learning rate. | $[10^{-6}, 10^{-3}]$ |
| *Batch size* | Batch size. | $\{32, 128\}$ |
| *Seq len* | Sequence length. | $\{10, 25, 50\}$ |
| *Embedding units* | Size of the embedding layer. | $\{400, 850\}$ |
| *Recurrent units* | Size of the recurrent layers. | $\{400, 850, 1150\}$ |
| *LSTM layers* | Number of recurrent layers. | $\{1, 2, 3\}$ |
| *Dual units** | Size of the *dual* layer. | $\{400, 850\}$ |
| *Embedding L2reg* | L2 regularization applied to the Embedding and output layers. | $\{0, 10^{-6}, 10^{-5}\}$ |
| *Rec. input L2reg* | L2 regularization applied to the input weights of the recurrent layer. | $\{0, 10^{-6}, 10^{-5}\}$ |
| *Rec. L2reg* | L2 regularization applied to the recurrent weights of the recurrent layer. | $\{0, 10^{-6}, 10^{-5}\}$ |
| *Activation L2reg* | L2 regularization applied to the recurrent layers output. | $\{0, 10^{-6}, 10^{-5}\}$ |
| *Dual L2reg** | L2 regularization applied to *dual* layer. | $\{0, 10^{-6}, 10^{-5}\}$ |
| *Rec. input Dropout* | Dropout before the first recurrent layer. | $[0.0, 0.5]$ |
| *Rec. Dropout* | Dropout for the linear transformation of the recurrent state. | $[0.0, 0.5]$ |
| *Rec. internal Dropout* | Dropout between the recurrent layers. | $[0.0, 0.5]$ |
| *Rec. output Dropout* | Dropout after the last recurrent layer. | $[0.0, 0.5]$ |
| *Dual input Dropout** | Dropout before the *dual* layer. | $[0.0, 0.5]$ |
| *Dual output Dropout** | Dropout after the *dual* layer. | $[0.0, 0.5]$ |
| *Mogrifier deep* | Mogrifier rounds. | $\{0, 2, 3, 4, 5, 6\}$ |
| *Mogrifier L2reg* | L2 regularization applied to Mogrifier weights. | $\{0, 10^{-6}, 10^{-5}\}$ |
| *Mogrifier rank* | Weight factorization. $\boldsymbol{Q}^i \in \mathbb{R}^{m \times n} = \boldsymbol{Q}_l^i \boldsymbol{Q}_r^i$ with $\boldsymbol{Q}_l^i \in \mathbb{R}^{m \times k}, \boldsymbol{Q}_r^i \in \mathbb{R}^{k \times n}$. | $\{0, 50, 100, 200\}$ |
| *Mogrifier Dropout* | Dropout between the Mogrifier weights. | $[0.0, 0.2]$ |
| *Learning rate eval* | Learning rate when Dynamic evaluation. | $[10^{-6}, 10^{-3}]$ |
| *Seq len eval* | Sequence length when Dynamic evaluation. | $[5, 50]$ |
| *Clipnorm eval* | Gradients clipping to a maximum norm. | $[0.0, 1.0]$ |

## 4 RESULTS

We first show the results of the comparative analysis ERS vs Dual, then we focus on the search of the optimal hyperparameters for the *dual* architecture with the $mdLSTM$ recurrence.

### 4.1 DUAL VS NON-DUAL ARCHITECTURES

Table 2 displays the validation and test perplexity scores obtained for each of the experimental configurations on the PTB and the WT2 problems, both with and without dynamic evaluation. To facilitate the comparison, each pair of rows contain the results for one of the recurrent modules ($LSTM$, $dLSTM$ or $mdLSTM$) using the two architectures ERS and Dual, with the best values shown in bold. In each case, the hyperparameters are tuned for the standard ERS architecture and then used within the *dual* networks without any additional adaptation. The exceptions are hyper-

Table 2: Validation and test word-level perplexity obtained for each of the experimental configurations on the PTB (top) and the WT2 (bottom) datasets.

**Penn Treebank Dataset**

| MODEL | No. PARAMS | No Dyneval | | Dyneval | |
|---|---|---|---|---|---|
| | | Val. | Test | Val. | Test |
| *LSTM* | 8.88 M | 67.37 | 64.91 | 62.31 | 61.17 |
| *Dual LSTM* | 9.60 M | 61.22 | 59.39 | **55.26** | **54.69** |
| *dLSTM* | 13.62 M | 63.44 | 61.03 | 57.18 | 56.01 |
| *Dual dLSTM* | 13.94 M | 60.99 | 59.56 | **56.11** | **54.87** |
| *mdLSTM* | 21.43 M | 57.42 | 55.48 | 51.16 | 50.27 |
| *Dual mdLSTM* | 22.88 M | 56.08 | 54.12 | **48.82** | **48.00** |
| *mdLSTM+* | 22.16 M | 57.77 | 56.29 | 50.42 | 49.83 |

**WikiText-2 Dataset**

| MODEL | No. PARAMS | No Dyneval | | Dyneval | |
|---|---|---|---|---|---|
| | | Val. | Test | Val. | Test |
| *LSTM* | 20.23 M | 92.84 | 88.28 | 74.98 | 69.42 |
| *Dual LSTM* | 20.95 M | 85.88 | 82.48 | **61.94** | **57.61** |
| *dLSTM* | 29.60 M | 78.65 | 75.60 | 63.26 | 59.42 |
| *Dual dLSTM* | 30.32 M | 77.01 | 73.90 | **61.10** | **57.10** |
| *mdLSTM* | 37.51 M | 72.05 | 69.06 | 57.42 | 53.93 |
| *Dual mdLSTM* | 38.95 M | 71.78 | 70.83 | **53.48** | **50.71** |

parameters, such as the *dual* dropout, which do not exist in the ERS configuration (those marked with an asterisk in table 1). To give a measure of the model complexity, table 2 contains also the approximate number of trainable parameters for each configuration.

As expected, dynamic evaluation improves the results regardless of the model or the dataset. The main observation, however, is that networks enhanced with the *dual* connection display lower perplexity scores for almost all the training conditions on both the PTB and the WT2 datasets. The advantage of the Dual vs the ERS architecture is larger for less complex models, and narrows as the model complexity increases. Nevertheless, even for networks with $mdLSTM$ recurrence, the *dual* architectures outperform their non-dual counterparts in more than 2 perplexity points on the test set, when dynamic evaluation is used.

In order to test that this improvement is due to the *dual* connection and not to the presence of an extra processing layer, we performed an additional experiment with a *Dual mdLSTM* model, but removing the term $W^{de}e_t$ from equation 6. The results for the PTB dataset are shown in table 2 as *mdLSTM+*. Note that, in spite of slightly improving the baseline, this enhanced mogrifier model is still well below the result obtained with the full *dual* architecture.

Finally, it is worth noting that all the results presented correspond to our own implementation of the models, and that in most cases we are not including some of the several training or validation adaptations frequently used in the literature (such as AWD or MoS, for example). This can explain the difference with respect to the results reported by Melis et al. (2020) for the Mogrifier-LSTM model. We would expect a further improvement of the results were these additional mechanisms implemented.

Table 3: Best validation and test word-level perplexity scores reported in the literature for the Penn Treebank dataset, with and without dynamic evaluation. Missing values in the last two columns correspond to works where the dynamic evaluation approach was not considered. The last row in the table displays the results obtained with our *Dual mdLSTM* network.

| MODEL | | No Dyneval | | Dyneval | |
| --- | --- | --- | --- | --- | --- |
| | | Val. | Test | Val. | Test |
| *AWD-LSTM* (Merity et al., 2018) | 24 M | 60.00 | 57.30 | - | - |
| *AWD-LSTM-DOC* (Takase et al., 2018) | 23 M | 54.12 | 52.38 | - | - |
| *AWD-LSTM* (Krause et al., 2018) | 24 M | 59.80 | 57.70 | 51.60 | 51.10 |
| *AWD-LSTM +PDR* (Brahma, 2019) | 24 M | 57.90 | 55.60 | 50.10 | 49.30 |
| *AWD-LSTM +MoS* (Yang et al., 2018) | 22 M | 56.54 | 54.44 | 48.33 | 47.69 |
| *AWD-LSTM +MoS +PDR* (Brahma, 2019) | 22 M | 56.20 | 53.80 | 48.00 | 47.30 |
| *AWD-LSTM-DOC x5* (Takase et al., 2018) | 185 M | 48.63 | 47.17 | - | - |
| *AWD-LSTM +MoS +FRAGE* (Gong et al., 2018) | 24 M | 55.52 | 53.51 | 47.38 | 46.54 |
| *AWD-LSTM +MoS +Adv* (Wang et al., 2019) | 22 M | 54.98 | 52.87 | 47.15 | 46.52 |
| *AWD-LSTM +MoS +Adv +PS* (Wang et al., 2019) | 22 M | 54.10 | 52.20 | 46.63 | 46.01 |
| *Mogrifier-LSTM* (Melis et al., 2020) | 24 M | 51.40 | 50.10 | 44.90 | 44.80 |
| ***Dual mdLSTM* - ours** | 23 M | 52.87 | 51.19 | 45.13 | **44.61** |

## 4.2 DUAL MOGRIFIER FINE TUNING

The second part of the experiments consists of searching for the best hyperparameters in the configuration that provided the smallest perplexity in the previous setup, that is the *Dual mdLSTM* architecture. We carry out this experiment with the PTB problem. After an extensive search (see table 1), the best performance is obtained with a model with 850 units in the embedding layer, 850 units in each of the mogrifier LSTM layers, and 850 units also in the *dual* layer. The input, recurrent, internal, and output dropout rates are all set to 0.5, the *dual* input and output dropout rates are set to 0.5 and 0.4, respectively, and the mogrifier dropout rate is set to 0.15. Both the embedding and the *dual* L2 regularization parameters are set to $10^{-5}$. The mogrifier number of rounds is set to 4, and the rank to 100. All the remaining hyperparameters are set to 0.

After the training phase, we continue with a fine tuning of some additional hyperparameters, using the validation data. First, we look for the best sequence length in the range $[5, 70]$, and then we fine-tune the softmax temperature in the range $[0.9, 1.3]$. When using dynamic evaluation, we also look for the best gradient clipping value (in the range $[0.0, 1.0]$) and, following Melis et al. (2020), we repeat the whole procedure with the $\beta_1$ parameter of the Nadam optimizer set to 0, which resembles the RMSProp optimizer without momentum. The results are shown in table 3, together with the top perplexity scores reported in the literature for the same problem.

The state-of-the-art is dominated by several variations of the AWD-LSTM network (Merity et al., 2018), the most common being the inclusion of a Mixture of Softmaxes (MoS) (Yang et al., 2018). Other add-ons include Direct Output Connection (DOC) (Takase et al., 2018), which is a generalization of MoS, Frequency Agnostic word Embedding (FRAGE) (Gong et al., 2018), Past Decode Regularization (PDR) (Brahma, 2019), or Partial Shuffling (PS) with Adversarial Training (Adv) (Wang et al., 2019). The mogrifier-LSTM described in section 2.2 combines many of these ideas with a mutual gating between the input and the hidden state vectors to obtain the best results reported in the literature for the PTB problem, among those obtained by networks that do not use additional data during the training phase. Compared with all these models, our current approach leads the ranking with a perplexity score of 44.61, even though most of the aforementioned tricks have not been considered.

## 5 DISCUSSION

In this work, we have presented a new network design for the Language Modeling task based on the *dual* network proposed by Oliva & Lago-Fernández (2021). This network adds a direct connection between the input and the output, skipping the recurrent module, and can be adapted to any of the

traditional Embedding-Recurrent-Softmax (ERS) models, opening the way to new approaches for this task. We have based our work on the Penn Treebank (Mikolov et al., 2010) and the WikiText-2 (Merity et al., 2017) datasets, comparing the ERS approach and its *dual* alternative. Regardless of the configuration, the *dual* version performs always better, even though it faces a slight disadvantage, since most of the hyperparameters are tuned using the ERS model. We can expect a much better performance if the complete set of hyperparameters is properly tuned for the *dual* network.

This is in fact the case for the second experiment, where a *Dual mdLSTM*, which includes a simplified version of the mogrifier LSTM (Melis et al., 2020) within a *dual* architecture, is fine tuned for the Penn Treebank dataset. After a thorough search of the hyperparameters space, we have found a network configuration that establishes a new state-of-the-art score for this problem. Interestingly, this new record has been obtained in spite of leaving aside many of the standard features used in most state-of-the-art approaches, such as the Averaged SGD Weight-Drop (*AWD*) (Merity et al., 2018) or the Mixture of Softmaxes (*MoS*) (Yang et al., 2018). The incorporation of these features into the *dual* architecture can be expected to further increase the model performance.

The *dual* architecture was firstly proposed as an alternative that reduces the computational load on the recurrent layer, letting it concentrate on modeling the temporal dependencies only. From a more abstract point of view, it has been argued that the dual architecture can be understood as a sort of Mealy machine, where the output explicitly depends on both the hidden state and the input (Oliva & Lago-Fernández, 2021). Our results show that this explicit dependence on the input can indeed lead to better performance on language modeling tasks. This emphasizes the importance of the current input in RNN models.

Finally, although the new approach has not been tested with large-scale language corpora, we expect that our results scale well to larger datasets. Work in progress contemplates this extension. The *dual* architecture also needs further research concerning the deepness of the specific variations of Language Modeling and other families of problems not necessarily related to Natural Language Processing. This work opens a new line of research to be considered when processing any sequence or time series. The utility of this approach in more general problems will be addressed in future work.

ACKNOWLEDGMENTS

Omitted to preserve anonymity.

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
