# OpenReview forum: "The Importance of the Current Input in Sequence Modeling"
_ICLR.cc/2022/Conference — ICLR 2022 Submitted_

### Official Review · Reviewer_zNdy · 2021-10-24

**Correctness:** 4
**Technical Novelty And Significance:** 2
**Empirical Novelty And Significance:** 2
**Recommendation:** 3
**Confidence:** 3

**Main Review:**

Strength:

It's good to revisit the LSTM recurrent structure.

Hyper-parameter tuning is reported in detail.

I got a bit surprised by the improvement from the dual structure.

Weakness:

The used datasets like PTB or WT2 are small-scale. I'd like to see something like wiki-103.

Why there's no comparison with transformers?

Not much analysis or explanation about why the dual structure is better is given.

Below are some minor comments:

I sometimes got confused by "dual" and "double" in the writing.

Some possible typos:

Sec1: "for the Penn Treebank problem "-> PTB "dataset"?

Sec2.2: "achieving the highest perplexity scores" -> lowest PPL?

**Summary Of The Paper:**

This work revisits the LSTM architecture. They propose to modify a recurrent architecture by adding a direct connection between the input and the output of the recurrent module, called "dual". They also consider a double-layer LSTM, where the output of the recurrent module is obtained by the concatenated application of two LSTM layers.
In experiments, the dual modification can get consistent improvement.

**Summary Of The Review:**

While I think this revisit to LSTM is interesting, the scope of the experiments are still limited. Comparison with transformers is not conducted, and the datasets are small scale.

---

> ### Author Response · Authors · 2021-11-21
> **Response**
>
> Point 1: The used datasets like PTB or WT2 are small-scale. I'd like to see something like wiki-103.
>
> The datasets we used are definitely on the small-scale side. Nevertheless, they are still used in many research works related to language modeling, and they are still useful to illustrate the advantages of the proposed approach. Our results on these datasets show that the dual architecture provides a consistent improvement over the baseline that is significant for all the experiments we carried out (table 2). Additionally, we slightly improve the state of the art for the PTB dataset when no external data are used to train the models (table 3). For all these reasons we consider that the use of small-scale datasets should not be a justification to downplay our approach. On the other hand, we agree that the use of larger datasets would be a bonus, as we already point out in the discussion section of the article, but this is obviously unaffordable during the revision period.
>
> Point 2: Why there's no comparison with transformers?
>
> The main reason is that transformers are models with a huge number of parameters (around 175 billion for GPT-3, for example) that need to be trained with external data. Transformer models trained with just the PTB data typically obtain a worse perplexity score than the mogrifier LSTM network that has been taken as a baseline in our research.
>
>
> Point 3: Not much analysis or explanation about why the dual structure is better is given.
>
> The advantage of the new architecture comes from the inclusion of an alternative path between the input and the output that skips all the recurrent modules. This combines the benefits of recurrent and pure feed-forward architectures. All the relevant information contained in the current input can then be processed by the feedforward layer, discharging the recurrent layers from part of the computational load and letting them focus on just the temporal dependencies. As we suggest in the discussion, if a standard recurrent neural network is idealized as a sort of finite state machine (in its Moore’s version), our new model can be imagined as a Mealy machine, where the output explicitly depends on both the state and the input. Mealy machines are typically simpler (in number of states) than Moore machines, so by reducing the complexity of the problem that the recurrency needs to solve we are simplifying the network’s task. If we pay attention to the results, the inclusion of the extra dual connection improves the perplexity score with respect to the non-dual baselines in all our experiments.
>
> Responses to some comments of the other reviewers also deepen into the mechanisms behind the observed improvement associated with the dual connection.
>
> Point 4: I sometimes got confused by "dual" and "double" in the writing.
>
> We use the term dual to refer to the new connection from the embedding layer to the output, skipping all the recurrent modules. The term double makes reference to the concatenation of two LSTM layers.
>
> Point 5: Some possible typos...
>
> We have corrected the typos in the reviewed version of the article.

---

> ### Comment · Reviewer_zNdy · 2021-11-29
> **Thanks for the response!**
>
> Thanks for the response!

---

### Official Review · Reviewer_6pt8 · 2021-11-02

**Correctness:** 3
**Technical Novelty And Significance:** 3
**Empirical Novelty And Significance:** 2
**Recommendation:** 5
**Confidence:** 5

**Main Review:**

I found the paper to be clearly written and easy to follow. This is an experimental paper, and the authors have taken care to describe the setup in sufficient detail. I also appreciated the use of two separate sets of experiments, one with hyperparameters for the baselines transferred over to the "dual" models, and another with hyperparameters tuned for the "dual" models.

There are a few concerns that make it difficult for me to accept this paper in its current form:

1. It is unclear where the improvement is coming from. Is the added dual connection improving ease of optimization, model capacity, or regularization (providing a useful inductive bias)? It's possible that this will be much clearer if the training and validation loss curves are compared for models in Table 2. Results for additional baseline models with slightly larger sizes might also help: those models should have slightly more capacity in principle, so how do their training and validation loss curves compare to the "dual" models.
2. Related to the above question: it appears that Melis et al. [A] use skip connections in their LSTMs to improve optimization, but this paper doesn't appear to use them. This appears to be very relevant here if the improvement from the proposed dual layer is coming from improved optimization. Would simpler skip connections provide a similar benefit? This can be checked by modifying the baseline to use additive skip connections.
3. Overall the experimental results are not strongly in favor of the proposed model. In Table 2, we find that dual layer leads to worse results without dynamic evaluation when added to "mdLSTM", but better results with dynamic evaluation. We don't know why this should be the case, and it casts doubt on a universal benefit of this design. We see a similar situation in Table 3 too, where the results are worse without dynamic eval, as well as with dynamic eval on the val set. Only on the test do we see a small improvement. It is very difficult to say from these results that the proposed design can be expected to improve performance in general. On this point, I don't have concrete suggestions to offer, but perhaps the authors can address this by adding experiments on more datasets, and/or clarify the reason for these mixed results.

If 1 & 2 are satisfactorily addressed, I'm willing to raise my score up to 6. If 3 can also be addressed, I will increase the score further.

[A] Melis, G., Dyer, C., & Blunsom, P. (2017). On the state of the art of evaluation in neural language models. arXiv preprint arXiv:1707.05589.

————
Update:
I am thankful to the authors for providing additional results and analysis. My concerns are partially addressed so I am increasing my score. However, I believe more work is needed to make sure that the increased capacity provided by the _dual_ connection can actually be useful in practice. What I mean by this is that many results in Table 2, while useful from a comparative perspective, feel less relevant from a practical perspective because they show that the _dual_ connection improves performance of models _underpowered for the task/dataset_ at hand. The most relevant results are when we get close to the “best” test set performance for the task. It is here the evidence gets fuzzy, and the use of only 2 datasets makes the issue difficult to ignore.
Can I take a setup that’s otherwise well tuned for a task, add this connection to improve capacity, and expect improvement? If not, why not, and under which conditions should I expect it? I don’t have sufficient confidence to answer these questions based on the results so far. The experiments suggested by the authors in the responses and in the paper’s Discussion section may be necessary to provide that confidence.

**Summary Of The Paper:**

This paper proposes a simple improvement to recurrent network architectures for language modeling. The idea is to insert a single additional layer into a network with one or more recurrent layers, just before the output layer (Eq. 6). This is termed a "dual connection" and it combines the output of the last recurrent layer directly with the input to the network at the current step. The paper presents the results of adding this modification to both LSTM and Mogrifier-LSTM networks on the Penn Treebank and WikiText-2 datasets.


**Summary Of The Review:**

The paper proposes a simple modification to recurrent language models that can potentially improve performance, but additional analysis and results are needed to establish the utility of the approach.

---

> ### Author Response · Authors · 2021-11-21
> **Response part 2**
>
> Point 2: Related to the above question: it appears that Melis et al. [A] use skip connections in their LSTMs to improve optimization, but this paper doesn't appear to use them...
>
> Skip connections are useful to improve optimization, but they do not in principle increase the capacity of the models. As we showed with our response to point 1, the benefit of the dual connection is more related to capacity than to optimization. Nevertheless, we found it very interesting the suggestion to check the models’ behavior also with skip connections. Hence we have performed two additional experiments where both the baseline and the dual models are enhanced with an additive connection that skips the second LSTM module. The results are shown in figures 3 and 4 of the supplementary material. We observe that additive skip connections seem to be relevant only for the baseline model, mdLSTM. They allow for a faster convergence, although no significant improvement is obtained in the long term for the validation data (figure 3). The use of skip connections within the dual model does not contribute any appreciable difference with respect to the standard Dual mdLSTM (figure 4), but there is a slight advantage in favor of the model without skip connections.
>
> Point 3: Overall the experimental results are not strongly in favor of the proposed model…
>
> The results displayed in table 2 are consistent and clearly in favor of the proposed approach. The reviewer is basing his argument on one single case out of the 24 experimental results shown in the table, namely the comparison between the test perplexity scores for the mdLSTM and the Dual mdLSTM models on the WikiText-2 dataset when no dynamic evaluation is used. For all the remaining situations the use of the dual model contributes a perplexity gain of at least two points (and much bigger in general). The deviation with respect to this generality highlighted by the reviewer could be due to a statistical fluctuation. It is also worth noting that the hyperparameters of all the models in table 2 have been tuned using the non-dual versions of the models. We would expect this anomaly to be corrected if we fine-tune the hyperparameters of the dual architecture.
>
> With respect to the results in table 3, we would like to comment on two important aspects. First, the last two rows in table 3 do not correspond to equivalent models, with and without the dual connection. The penultimate row corresponds to the results reported by Melis et al., 2020, with their Mogrifier-LSTM model. The last row contains the results of our dual mdLSTM model, which uses the mogrifier transformation but lacks many of the additional characteristics of the Melis et al. model. So, even though the fine-tuned Dual mdLSTM model obtains a gain of more than 5 perplexity points with respect to its baseline (mdLSTM in table 2), the overall results are similar to those obtained by Melis et al., with a slight advantage towards our approach on the test set and towards theirs on the validation set. These differences are probably not statistically significant. It is important to emphasize that the starting point for the dual model is not the Melis et al. model but the simpler mdLSTM model whose perplexity score is well below the former’s (50.27 vs 44.80). In spite of this initial disadvantage, the inclusion of the dual connection is sufficient for the model to slightly surpass the best results reported in the literature (for models that do not use external data) on the test set. We expect that these results can be further improved by using the dual connection within the original Mogrifier-LSTM model, but we have not been able to test this point.
>
> Second, due to limitations in our computational power, the hyperparameter optimization has been performed only for the dual mdLSTM model under the dynamic evaluation condition. The optimal hyperparameters found this way are then used to train the model without dynamic evaluation. This might explain the loss of performance with respect to the Mogrifier LSTM when no dynamic evaluation is used.

---

> ### Author Response · Authors · 2021-11-21
> **Response part 1**
>
> We would like to thank the anonymous reviewer for the thorough revision and the very interesting suggestions. We have tried to address all the concerns, as we explain in the following points.
>
> Point 1: It is unclear where the improvement is coming from…
>
> Following the reviewer’s suggestion, we have analyzed the loss curves comparing the mdLSTM (baseline) and the Dual mdLSTM models, and we have performed additional experiments where the baseline capacity is increased by adding more units to the recurrent layers. The loss curves have been added to our submission as supplementary material (figures 1 and 2).
>
> First, from figure 1 we observe that both the baseline and the dual models have a similar convergence ratio, but the dual model has a higher capacity. Additionally, we have also conducted a new experiment to check where this extra capacity comes from (if it is due to the new dual connection or just to the extra feed-forward layer in equation 6). Hence, we have run a new set of experiments where we remove the input-output connection but keep the extra layer. That is, we substitute equation (6) by d_t = ReLU(W_dh h_t + b_d). The results, which have been added to table 2 for the PTB dataset, show that the extra feedforward layer is not enough to explain the perplexity gain observed when using the dual architecture (see also response to point 2 from reviewer 1).
>
> Second, in figure 2 we compare the performance of three different models: the baseline (mdLSTM, 21.43 M parameters), the dual model (Dual mdLSTM, 22.88 M parameters), and an increased baseline (BIG mdLSTM, 53.42M parameters), which is an mdLSTM where the number of recurrent units has been increased from 850 to 1500 in the two LSTM layers. The main observation, in this case, is that the extra capacity in the increased baseline leads to more overfitting, whereas the dual model, in spite of a larger training loss, is able to obtain a better validation performance by the end of the training phase. Regarding the perplexity score, the values for the test set after 100 training epochs are 50.27, 49.61, and 48.00 for the mdLSTM, the BIG mdLSTM, and the Dual mdLSTM models respectively.

---

### Official Review · Reviewer_m9iZ · 2021-11-02

**Correctness:** 3
**Technical Novelty And Significance:** 1
**Empirical Novelty And Significance:** Not applicable
**Recommendation:** 3
**Confidence:** 5

**Main Review:**

Strengths:
+ The technique is very simple and is applicable to any embedding-recurrent-softmax (ERS) architecture.
+ The presentation of the method is very clear. Sufficient details were given on the experiment setup and hyperparameter search.

Weaknesses:
+ The proposed technique, while simple, is not very novel. A very similar technique has been examined in a previous paper (https://ieeexplore.ieee.org/document/9207238), and similar experiments on PTB have also been conducted.
+ It's hard to justify whether the performance gain in the Table 1 actually due to the residual connections or because of the extra input projections introduced. The authors could have examined this by simply doing a version of the residual connection by directly adding the input embedding $x_t$ to the hidden state $h_t$, which will incur no extra parameters at all.
+ After hyperparameter tuning, the performance gain brought by the architectural change becomes very marginal.

I also have a question for the author: why is the parameter number for Dual mdLSTM smaller than Mogrifier-LSTM in Table 3?

=== UPDATE ===

I have read the authors response and the other reviewers' reviews and do not think any change in my overall evaluation is justified. See my detailed comment below.

**Summary Of The Paper:**

This paper proposes to add an extra residual connection between a transformed word embedding and the final output layer, which easily generalizes over different recurrent architectures. The language model experiments show that their proposed model performs better than the same recurrent model without the residual connection, but the performance gain becomes marginal after optimal hyperparameter setups were used.

**Summary Of The Review:**

The idea is simple and generally applicable, but the lack of novelty and the marginal result improvement greatly limit the impact of the paper.

---

> ### Author Response · Authors · 2021-11-21
> **Response part 1**
>
> We thank the reviewer for the interesting comments and suggestions that can help to improve our work. In the following we provide a detailed response to all the raised concerns.
>
> Point 1: The proposed technique, while simple, is not very novel...
>
> After carefully reading the article mentioned by the reviewer, we have to disagree on this point. The article by Kuo et al. introduces a modification of the LSTM, GRU, and other gated RNN architectures by using an input residual connection (IRC) that alleviates the design and considerably reduces the number of parameters. This modification alters the recurrent module, but there is no connection in their networks that directly links the input layer to the output layer skipping the recurrence. So, even with the new IRC, their model is still a standard Embedding-Recurrent-Softmax (ERS) architecture. In our approach, however, we do not alter the recurrent equations, but include an alternative path between input and output that skips all the recurrent modules. Hence we are combining the benefits of recurrent and pure feed-forward architectures. If we pay attention to the results, their best perplexity score on the Penn Treebank dataset is more than 10 points below ours.
>
> Point 2: It's hard to justify whether the performance gain in the Table 1 actually due to the residual connections or because of the extra input projections introduced…
>
> We assume that the reviewer refers to the results shown in table 2. This is a very interesting point that is worth clarifying. First, even if we directly connect the embedding layer to the softmax output we have an increase in the number of parameters since the embedding vector e and the recurrent output h are combined using concatenation, and not addition. Second, the direct connection from embedding to softmax would be of little use here because the softmax layer is tied to the embedding. Despite these two facts, we agree that it is necessary to understand whether the observed improvement is due to the new input-output connection and not to the extra feedforward layer. So we have conducted a new set of experiments where we remove the input-output connection but keep the extra layer. That is, we substitute equation (6) by d_t = ReLU(W_dh h_t + b_d). The results, which have been added to table 2 for the PTB dataset, show that the extra feedforward layer is not enough to explain the perplexity gain observed when using the dual architecture.
>
> Please also note a change in the perplexity values shown in table 2 for the mdLSTM and the Dual mdLSTM models on the PTB data. Due to an error, in the previous version of the article we were using values after 200 training epochs, so the comparison with the remaining experiments, which were run for just 100 epochs, was unfair. This change does not alter our main message.

---

> ### Author Response · Authors · 2021-11-21
> **Response part 2**
>
> Point 3: After hyperparameter tuning, the performance gain brought by the architectural change becomes very marginal.
>
> There is a possible misinterpretation of the results presented in tables 2 and 3. Observing table 2, it is clear that the performance gain associated with the new architecture is not marginal at all. We observe a consistent improvement of the perplexity score for all the experimental conditions when the dual connection is added to the networks. In all cases, the dual architecture outperforms its non-dual counterpart in more than 2 perplexity points, even when the hyperparameters have been tuned with the non-dual networks.
>
> In table 3 we fine-tune the hyperparameters of our best model in order to compare our results with the state of the art. The confusion may come from assuming that the last two rows in table 3 correspond to equivalent models, with and without the dual connection. But this is not true. The penultimate row corresponds to the results reported by Melis et al., 2020, with their Mogrifier-LSTM model. The last row contains the results of our dual mdLSTM model, which uses the mogrifier transformation but lacks many of the additional characteristics of the Melis et al. model. So, regarding the additional improvement associated with fine-tuning the mdLSTM model, the fair comparison would be with the result shown in table 2. In this case, we observe an additional gain of more than 5 perplexity points when the hyperparameters are fine-tuned to the dual architecture. Finally, it is true that our improvement with respect to the state of the art results could be considered marginal, but one should note that the starting point is not the Melis et al. model but the simpler mdLSTM model whose perplexity score is clearly below the former’s (50.27 vs 44.80). In spite of this initial disadvantage, the inclusion of the dual connection is sufficient for the model to slightly surpass the best results reported in the literature (for models that do not use external data). We expect that these results can be further improved by using the dual connection within the original Mogrifier-LSTM model, but we have not been able to test this point.
>
> Point 4: I also have a question for the author: why is the parameter number for Dual mdLSTM smaller than Mogrifier-LSTM in Table 3?
>
> As we explained previously, the Dual mdLSTM model in the last row of table 3 is not exactly the Mogrifier-LSTM model by Melis et al. augmented with a dual connection. The starting point for the Dual mdLSTM is the mdLSTM, a simplified version of the Mogrifier-LSTM with a smaller number of parameters (see table 2). This explains that, in spite of adding additional weights associated with the dual connection, the number of parameters of the Dual mdLSTM is still smaller than the Mogrifier-LSTM.

---

> > ### Comment · Reviewer_m9iZ · 2021-11-27
> > **Reviewer Comment After Response**
> >
> > Thanks the authors for the detailed response.
> >
> > **Point 1**: "If we pay attention to the results, their best perplexity score on the Penn Treebank dataset is more than 10 points below ours."
> >
> > I assume the authors mean 10 points above because lower PPL is better? That being set aside, comparing the best result from these two papers is unfair because Kuo et al. did not use dynamic evaluation.
> >
> > Also, the key idea I'm trying to convey in this point is that -- this is a small architectural modification from the start, and made even smaller by some existing literature that is similar. As a result, unless there is clear advantage that the proposed modification is better than the baseline and other variants of the similar idea, I don't think this will make a lot of impact. I think the other reviewers are expressing a similar concern and I don't think the response from the author satisfactorily addresses it.
> >
> > **Point 2**: Thanks for producing the additional results. Although I don't think this really fully addressed my concerns -- basically this architectural modification introduced two additional projections, namely $W^{de}$ and $W^{dh}$. If I understood correctly, what the author did was to get rid of $W^{de}$, but the extra parameters $W^{dh}$ are still there. So naturally, less extra parameters will give less improvements, which is what the experiments are showing.
> >
> > What I had in mind was to force $x_t$ and $h_t$ of the same dimension, and then the author could test if directly passing $x_t + h_t$ to the final layer is helpful vs. passing $h_t$ (in this way you can actually do element-wise add instead of concatenation). Of course this will not be comparable with the rest of the results because of the differences in hyper-parameter choices, but I think this will give the readers a clearer idea as to the necessity of this residual connection. I also realize that my original description was possibly not clear, apologies for that.
> >
> > **Point 3 and 4**: This is clearer now, but was very obscure in the original version of the paper. However, seeing this response, I also think that the point "we expect that these results can be further improved by using the dual connection within the original Mogrifier-LSTM model" is actually a crucial one to convince the readers that this architectural modification is empirically useful.
> >
> > Hence, at the end, unfortunately I don't think the response warrant a change to my review. I intend to keep my original evaluation.

---

### Decision · Program_Chairs · 2022-01-20

**Decision:**

Reject

**Comment:**

This paper considers augmenting LSTM language models with a form of residual connection that adds and additional feed forward layer before the softmax that integrates the output of the recurrent cell with the input embedding. This architectural variation is evaluated on the standard Penn Treebank and Wikitext-2 language modelling tasks and shown to lead to lower perplexities on the test sets, particularly when dynamic evaluation is used.

The reviewers agree that the proposed addition is well motivated, however they also observe that there has been substantial work in language modelling on various forms of residual and skip connections and it is not clear how this work relates to that body of work. The authors have provided some additional comparisons during the discussion, however the reviewers feel that further evaluation and analysis is needed. There was also some additional confusion about the varying hyperparamter tuning protocols employed in the different evaluations. The author’s have clarified this in their response so that it is clearer how the different results were obtained.

Overall this paper presents an promising initial result, but it would benefit from more complete evaluation, analysis, and hyperparameter tuning. This could include ablation studies and analysis to shed more light on what the proposed architectural addition is contributing, how this relates to other varieties of residual connection, and it’s positive interaction with dynamic evaluation. It would also be useful to include a tuned model with a comparison to previously reported Wikitext-2 results.